

# A comparative study of machine learning and deep learning algorithms to classify cancer types based on microarray gene expression data

Reinel Tabares-Soto[1], Simon Orozco-Arias[2,3], Victor Romero-Cano[4], Vanesa Segovia Bucheli[5], José Luis Rodríguez-Sotelo[1] and Cristian Felipe Jiménez-Varón[6]

[1] Department of Electronics and Automation, Universidad Autónoma de Manizales, Manizales, Caldas, Colombia
[2] Department of Computer Science, Universidad Autónoma de Manizales, Manizales, Caldas, Colombia
[3] Department of Systems and informatics, Universidad de Caldas, Manizales, Caldas, Colombia
[4] Department of Automatics and Electronics, Universidad Autónoma de Occidente, Cali, Valle del Cauca, Colombia
[5] İzmir International Biomedicine and Genome Institute, Dokuz Eylül University, Izmir, Turkey
[6] Department of Physics and Mathematics, Universidad Autónoma de Manizales, Manizales, Caldas, Colombia

Corresponding authors
Reinel Tabares-Soto,
rtabares@autonoma.edu.co
Simon Orozco-Arias,
simon.orozco.arias@gmail.com

## ABSTRACT

Cancer classification is a topic of major interest in medicine since it allows accurate and efficient diagnosis and facilitates a successful outcome in medical treatments. Previous studies have classified human tumors using a large-scale RNA profiling and supervised Machine Learning (ML) algorithms to construct a molecular-based classification of carcinoma cells from breast, bladder, adenocarcinoma, colorectal, gastro esophagus, kidney, liver, lung, ovarian, pancreas, and prostate tumors. These datasets are collectively known as the 11_tumor database, although this database has been used in several works in the ML field, no comparative studies of different algorithms can be found in the literature. On the other hand, advances in both hardware and software technologies have fostered considerable improvements in the precision of solutions that use ML, such as Deep Learning (DL). In this study, we compare the most widely used algorithms in classical ML and DL to classify the tumors described in the 11_tumor database. We obtained tumor identification accuracies between 90.6% (Logistic Regression) and 94.43% (Convolutional Neural Networks) using $k$-fold cross-validation. Also, we show how a tuning process may or may not significantly improve algorithms' accuracies. Our results demonstrate an efficient and accurate classification method based on gene expression (microarray data) and ML/DL algorithms, which facilitates tumor type prediction in a multi-cancer-type scenario.

## INTRODUCTION

Cancer is one of the most deadly diseases in human health caused by the abnormal proliferation of cells, leading to malignant malformations or tumors with different pathology characteristics (*Varadhachary, 2007*). Cancer-type classification is critical to increasing patient survival rates. Molecular genetic analyses have discovered genetic alterations, or signatures, with different biological characteristics that allow discerning the responses to several treatments (*Greller & Tobin, 1999*). This enables early diagnosis and an accurate treatment; therefore, ensuring the efficacy and reduction of side effects (toxicity) of the treatment (*Wang et al., 2005*).

Impaired gene expression is a characteristic of carcinogenic cells (*Su et al., 2001*). Accordingly, microarray gene expression data from tumor cells provide an important source of information to improve cancer diagnosis in a cost-efficient manner, allowing the use of this strategy in developing countries. Since microarray datasets contain thousands of different genes to be analyzed, an accurate and efficient way of analyzing this amount of data is by Machine Learning (ML) and Deep Learning (DL) algorithms (*Motieghader et al., 2017*). In particular, these algorithms have been applied in other biological areas, including rules of association (*Orozco-Arias et al., 2019b*). Previous studies demonstrate the use of ML and DL in microarray gene expression to infer the expression of target genes based on landmark gene expression (*Chen et al., 2016*), in feature selection aimed at finding an informative subset of gene expression (*Sharma, Imoto & Miyano, 2012*), and in the diagnosis and classification of cancer types (*Fakoor et al., 2013*).

A well-known database of gene microarrays related to cancer is the 11_Tumors database (*Su et al., 2001*), which is available at https://github.com/simonorozcoarias/ML_DL_microArrays/blob/master/data11tumors2.csv. This dataset is a good example of the curse of dimensionality due to the high number of characteristics and few registers of this database. Therefore, most studies use it to test specific data science techniques, such as feature selection methods (*Bolón-Canedo et al., 2014*; *Wang & Wei, 2017*; *Han & Kim, 2018*; *Perera, Chan & Karunasekera, 2018*), dimension reduction (*Araújo et al., 2011*), clustering methods (*Sardana & Agrawal, 2012*; *Sirinukunwattana et al., 2013*; *Li et al., 2017*), preprocessing techniques (*Liu et al., 2019*), among others. The 11_Tumors database has also been used in gene selection for cancer classification (*Moosa et al., 2016*; *Alanni et al., 2019*). Although the authors achieved high accuracy in these publications, they only used some ML algorithms, one preprocessing strategy, and one learning technique (supervised or unsupervised), which could add bias to their methodology. Additionally, to date, no comparative study on the application of ML in microarray datasets is found in the literature.

In several ML studies, DL has proven to be a robust technique for analyzing large-scale datasets (*Bengio, Courville & Vincent, 2013*). With these advances, DL has achieved cutting-edge performance in a wide range of applications, including bioinformatics and genomics (*Min, Lee & Yoon, 2016*; *Yue & Wang, 2018*), analysis of metagenomics samples (*Ceballos et al., 2019*), identification of somatic transposable elements in ovarian cancer (*Tang et al., 2017*), identification and classification of retrotransposons in plants

(*Orozco-Arias, Isaza & Guyot, 2019*) and cancer classification using Principal Component Analysis (PCA) (*Liu, Cai & Shao, 2011*). Recent work by *Guillen & Ebalunode (2016)* demonstrated promising results for the application of DL in microarray gene expression.

In general, there are two different tasks that ML algorithms can tackle: supervised and unsupervised learning. In supervised learning, the goal is to predict the label (classification) or response (regression) of each data point by using a provided set of labeled training examples. In unsupervised learning, such as clustering and principal component analysis, the goal is to learn inherent patterns within the data (*Zou et al., 2018*).

The main goal of any ML task is to optimize model performance not only on the training data but also on additional datasets. When a learned model displays this behavior, it is considered to generalize well. With this aim, the data in a given database are randomly split into at least two subsets: training and validation (*Zou et al., 2018*). Then, a model as complex as possible is learned (training set), tuned (validation set), and tested for generalization performance on the validation set. This process is crucial for avoiding overfitting or underfitting. Therefore, a sound learning algorithm must reach an appropriate balance between model flexibility and the amount of training data. An overly simple model will underfit and make inadequate predictions, whereas an overly flexible model will overfit to spurious patterns in the training data and not generalize (*Zou et al., 2018*).

In this study, we compare the performance of the most commonly used ML and DL algorithms in bioinformatics (*Orozco-Arias et al., 2019a*) in the task of classifying by supervised and unsupervised techniques. We used the 11_Tumor database and applied different preprocessing strategies. Our detailed evaluation and comparison illustrate the high accuracy of these algorithms for tumor identification in a multiple-cancer-type scenario and the influence of preprocessing strategies and tuning processes on these accuracies.

## MATERIALS AND METHODS

ML and DL techniques can learn the characteristics of a given problem from a certain amount of data. These data are usually randomly subdivided into two groups: training and validation. A training dataset is used to calibrate the parameters of the model, and a validation dataset is utilized for evaluating model performance (*Eraslan et al., 2019*).

In this article, we compared results obtained from classifying 11 different tumor classes through different approaches of ML and DL. We began by evaluating two unsupervised methods; the first method is the popular *K*-means algorithm, in which a given number of prototype samples, also known as cluster centers, are estimated by iteratively assigning data points to prototype samples and updating them as the mean of the assigned samples. The second method tested is hierarchical clustering, which is better suited for irregular shapes than *K*-means. After, we tested eight different classification algorithms. The most popular one, and the standard baseline in classification problems, is K-Nearest Neighbors (KNN), where classification decisions are done through a voting mechanism and model training stores the dataset in a way that queries can be done efficiently. Another family of classification methods comprises the so-called linear models,

for which a learning algorithm estimates as many weights as features from the training data so classification prediction is done as a function of the dot product between the weights and a test sample. Linear models are fast to train, fast to predict, and also scale well to datasets in which the number of features is large compared to the number of samples. The linear methods we tested are Linear Support Vector Classifier (SVC), Logistic Regression (LR), Linear Discriminant Analysis (LDA), Naive Bayesian Classifier (NB), and Multi-Layer Perceptron (MLP).

We also included Decision Tree-methods (DT) such as Random Forests (RF). Unlike linear models, DTs and RFs are invariant to data scaling and work well with features on different scales. Finally, we applied Deep Neuronal Networks (DNN), such as fully connected neural networks, also known as Multi-Layer Perceptron (MLP) and Convolutional Neural Networks (CNNs). MLPs are well-suited for non-linear data, whereas CNNs automatize the costly task of engineering features; an unavoidable task in classical ML approaches. The above algorithms are extensively explained in *Michie, Spiegelhalter & Taylor (1994)* and *Chollet (2007)*.

## Datasets

The datasets used represent measurements of gene expression using cancer microarrays and normal biopsies (*Statnikov et al., 2005*; *Bolón-Canedo et al., 2014*), and are consolidated in the "11 Tumors database", which is freely available online at (https:// github.com/simonorozcoarias/ML_DL_microArrays/blob/master/data11tumors2.csv). This database consists of 174 samples with 12,533 gene expression microarrays for 11 different types of cancer. The 12,533 microarrays of genetic expression are integers with positive and negative values; these values represent the characteristics that allow the ML and DL algorithms to learn how to classify by cancer type. The types of cancer and the number of patients for each type are shown in Table 1. The classes of each cancer type are unbalanced and remained so in the experimentation.

## Preparing the data

For the experiments, we divided the information into two groups; the first group corresponds to the features (X) and the second group to the classes (Y). The features compose a matrix of size $m \times n$ and the classes are a vector of size $n \times 1$, where m is the number of samples and $n$ is the number of genes for each class (12,533). The dataset, containing 174 samples, is randomly subdivided into two subsets (80% training and 20% validation), including 139 samples for training and 35 samples for validation. Initial calibration of ML and DL algorithms (training) was done using the training set; then, hyperparameter tuning was performed with the validation set and measured the accuracy of the algorithms. We calculated the accuracy of each algorithm using tuned hyperparameters with $k$-fold cross-validation and $k = 10$ to avoid overfitting.

The dataset used in this paper has the curse of dimensionality since the number of characteristics (12,533) is higher than the number of samples (174) (*Powell, 2007*). Therefore, the data are dispersed and the results are not statistically stable or reliable, directly affecting the accuracy achieved by ML and DL algorithms. Two preprocessing

| Table 1 Cancer type classification in the 11_tumor database. | | |
|---|---|---|
| **Class** | **Cancer type** | **Number of patients** |
| 0 | Ovary | 27 |
| 1 | Bladder/Ureter | 8 |
| 2 | Breast | 26 |
| 3 | Colorectal | 23 |
| 4 | Gastroesophagus | 12 |
| 5 | Kidney | 11 |
| 6 | Liver | 7 |
| 7 | Prostate | 26 |
| 8 | Pancreas | 6 |
| 9 | Adenocarcinoma | 14 |
| 10 | Lung squamous cell carcinoma | 14 |

techniques were used to solve this problem: scaling (*Géron, 2017*) and principal component analysis (PCA) (*Wold, Esbensen & Geladi, 1987*). The first technique guarantees that the data are in a range of suitable values to calibrate the model. With the second technique, the statistical significance is improved and the noise introduced by irrelevant characteristics during model training decreases. In this paper, we worked with several combinations of the preprocessing techniques mentioned above to find the best performance.

Four different datasets were created for the training and validation of each ML or DL algorithm. For the first dataset, we did not apply any preprocessing operations; for the second, we performed a scaling process; for the third, we applied PCA with a retained variance of 96% to reduce data dimensionality, obtaining a dimensional reduction from 12,533 to 83 features. Finally, for the last dataset, we applied both scaling and PCA, obtaining a dimensional reduction from 12,533 to 113 features (principal components).

## Unsupervised learning experiments

Classification performance is highly correlated with the degree of separability of a dataset; therefore, we analyzed performance using clustering techniques. Based on data labels, we can gain a priori insight into the algorithm that works best on the distribution of the gene expression microarray dataset.

Before applying the classification algorithms (supervised learning), we performed a hierarchical analysis to better understand the dataset. This hierarchical clustering used different distance metrics, such as ward, average, single, complete, weighted, centroid, and median. Further, as input, we used a dataset with no preprocessing. These distance metrics serve to capture the differences between the data samples and vary in their capacity to deal with large outliers (i.e., between weighted, centroid, and median metrics) or if they allow choosing the number of clusters to consider (e.g., Ward) (*Foss, Markatou & Ray, 2019*). After this clustering, we tested all of the datasets created in the previous step to determine the best preprocessing methodology. Finally, a dendrogram and a heatmap

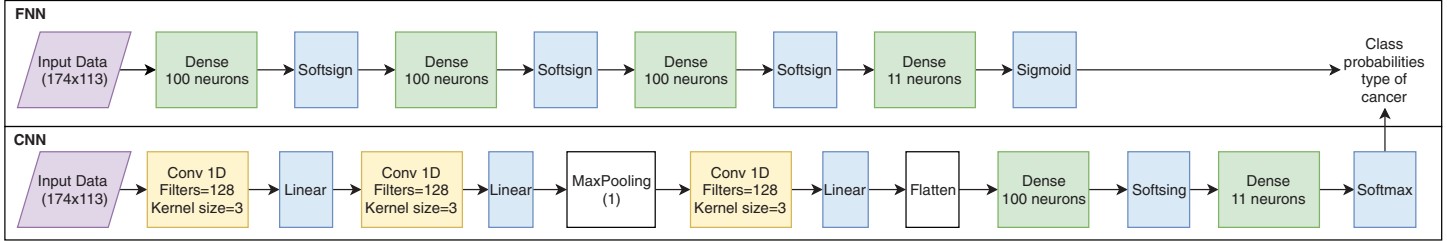

**Figure 1** Artificial neural network architectures used for cancer classification.

were used to illustrate the separability attribute of our dataset. Additionally, we performed a clustering analysis using the *K*-means algorithm with *k* values of one to eleven clusters using all datasets. We plotted the behavior in terms of accuracy and as a confusion matrix.

## Supervised learning

We evaluated the performance of well-known ML classification algorithms, including KNN, SVC, LR, LDA, NB, MLP, RF and DT. Subsequently, we evaluated DL architectures, such as fully connected neural networks (FNNs) and convolutional neural networks (CNNs).

## Neural network architecture

Two types of networks were used for DL; the first is a fully connected neural network and the second is a convolutional neural network. The FNN consists of three fully connected layers of 100 neurons each and the Softsign activation function; then, a final layer of 11 neurons is generated with the Sigmoid activation function to generate the probability of the type of cancer. The CNN consists of three convolutional layers with 128 filters each, with a kernel size of 3 and a linear activation function; followed by a layer of 100 fully connected neurons with the Softsign activation function and, finally, a layer of 11 neurons with the Softmax activation function to generate the probability of the type of cancer. Figure 1 shows the architectures used for the experiment, in which the top scheme is a FNN and the bottom scheme is a CNN.

## Tuning the algorithms

Several algorithms were tested by varying or tuning parameter values to find the best performance (Table 2). With these results, we plotted the accuracy values using all datasets created in the training and validation processes and also created confusion matrices. Finally, we did a cross-validation of each algorithm to find the accuracy that was less affected by bias. Additionally, in FNNs and CNNs, we performed a hyperparameter search with a grid search method (GridSearchCV) from the sklearn module, considering the variables shown in Table 3. Due to the high number of parameters, the process of tuning FNNs and CNNs involved choosing the parameter values that achieved the best accuracy and, then, using these values to find others. The process of finding the best parameter values is presented as follows: (1) batch size and epochs (2) training optimization

**Table 2 Tested algorithm parameters.**

| Algorithm | Parameter | Range | Step | Description |
|---|---|---|---|---|
| KNN | n_neighbors | 1–99 | 1 | Number of neighbors |
| SVC | C, gamma | C: 10–100, gamma: 1e−9 to 1e−4 | C:10, gamma: 10 | Penalty parameter C of the error term. Gamma is the free parameter of the Gaussian radial basis function |
| LG | C | 0.1–1 | 0.1 | Inverse of regularization strength |
| LDA | N/A | N/A | N/A | N/A |
| NB | N/A | N/A | N/A | N/A |
| MLP | solver='lbfgs', alpha=0.5, hidden_layer_sizes | 50–1,050 | 50 | Number of neurons in hidden layers. In this study we used solver lbfgs and alpha 0.5 |
| RF | n_estimators, max_depth, min_samples_split, max_features | n_estimators: 1–91, max_depth: 1–91, min_samples_split: 10–100, max_features: 10–90 | 10 for all parameters | N/A |
| DT | max_depth, min_samples_split, max_features | max_depth: 1–91, min_samples_split: 10–100, max_features: 10–90 | 10 for all parameters | N/A |
| K-means | n_clusters, random_state=0 | 1–17 | 1 | Number of clusters. In this study we used random state equals to zero |

algorithm (3) learning rate and momentum (4) network weight initialization (5) neuron activation function (6) dropout regularization and (7) number of neurons in the hidden layers.

## Significance tests

We performed a test for difference in proportions to determine whether the difference between accuracies of the algorithms is significant. We calculated the differences between the observed and expected accuracies under the assumption of a normal distribution. Given the number of correct test predictions $x$ and the number of test instances $N$, accuracy is defined as follows:

$$\text{Acc}_i = \frac{x}{N}$$

$$H_0 : \text{Acc}_i - \text{Acc}_j = 0$$

$$H_1 : \text{Acc}_i - \text{Acc}_j \neq 0$$

This test allowed determining if the accuracies of the algorithm change significantly after the tuning process and also if there are significant differences between the two algorithms with the highest average accuracies. Based on this, we evaluated whether the parameter tuning of the algorithms was necessary or if the ML algorithm used was more relevant.

**Table 3 Parameters tuned in DNNs.**

| Parameter | Values | Description |
|---|---|---|
| Batch size | 10, 20, 30, 40, 50, 60, 70, 80, 90, 100 | Number of training examples utilized in one iteration |
| Epochs | 10, 50, 100, 200 | Number of times that the learning algorithm will work through the entire training |
| Training optimization algorithm | SGD, RMSprop, Adagrad, Adadelta, Adam, Adamax, Nadam | Tools that update model parameters and minimize the value of the loss function, as evaluated on the training set |
| Learning rate | 0.001, 0.01, 0.1, 0.2, 0.3 | Hyper-parameter that controls how much the weights are being adjusting with respect to the loss gradient |
| Momentum | 0.0, 0.2, 0.4, 0.6, 0.8, 0.9 | Value between 0 and 1 that increases the size of the steps taken towards the minimum by trying to jump from a local minima |
| Network weight initialization | uniform, lecun_uniform, normal, zero, glorot_normal, glorot_uniform, he_normal, he_uniform | Initialization of weights into hidden layers of the network |
| Neuron activation function | softmax, softplus, softsign, relu, tanh, sigmoid, hard_sigmoid, linear | How the neuron output is activated based on its inputs |
| Dropout regularization | 0.0, 0.1, 0.2, 0.3, 0.4, 0.5, 0.6, 0.7, 0.8, 0.9 | Process of randomly dropping out nodes during training |
| Weight constraint | 1, 2, 3, 4, 5 | Value that introduces a penalty to the loss function when training a neural network to encourage the network to use small weights |
| Number of neurons in the hidden layers | 1, 5, 10, 20, 30, 40, 50, 60, 70, 80, 90, 100 | Amount of neurons that composed each hidden layers of the network |

## Tools

The algorithms were executed using Python programing language and scikit-learn libraries (*Pedregosa et al., 2011*), which are explained in *Komer, Bergstra & Eliasmith (2014)* for ML algorithms. PCA transformations and scaling were executed with the decomposition and preprocessing modules from scikit-learn. Also, DNNs were implemented using Keras (*Chollet, 2015*). All images were created with matplotlib (*Hunter, 2007*). The significance tests were performed using R software (Supplemental Material 1). The algorithms used here are available at https://github.com/simonorozcoarias/ML_DL_microArrays.

## RESULTS

### Hierarchical analysis

Before evaluating the classification algorithms, we visualized the intrinsic groupings in the data and determined how these groups are influenced by the different preprocessing methodologies applied to our data (Fig. 2). Using the downloaded raw data, we created a hierarchical graph (unsupervised learning) using different methodologies (Fig. S1) and

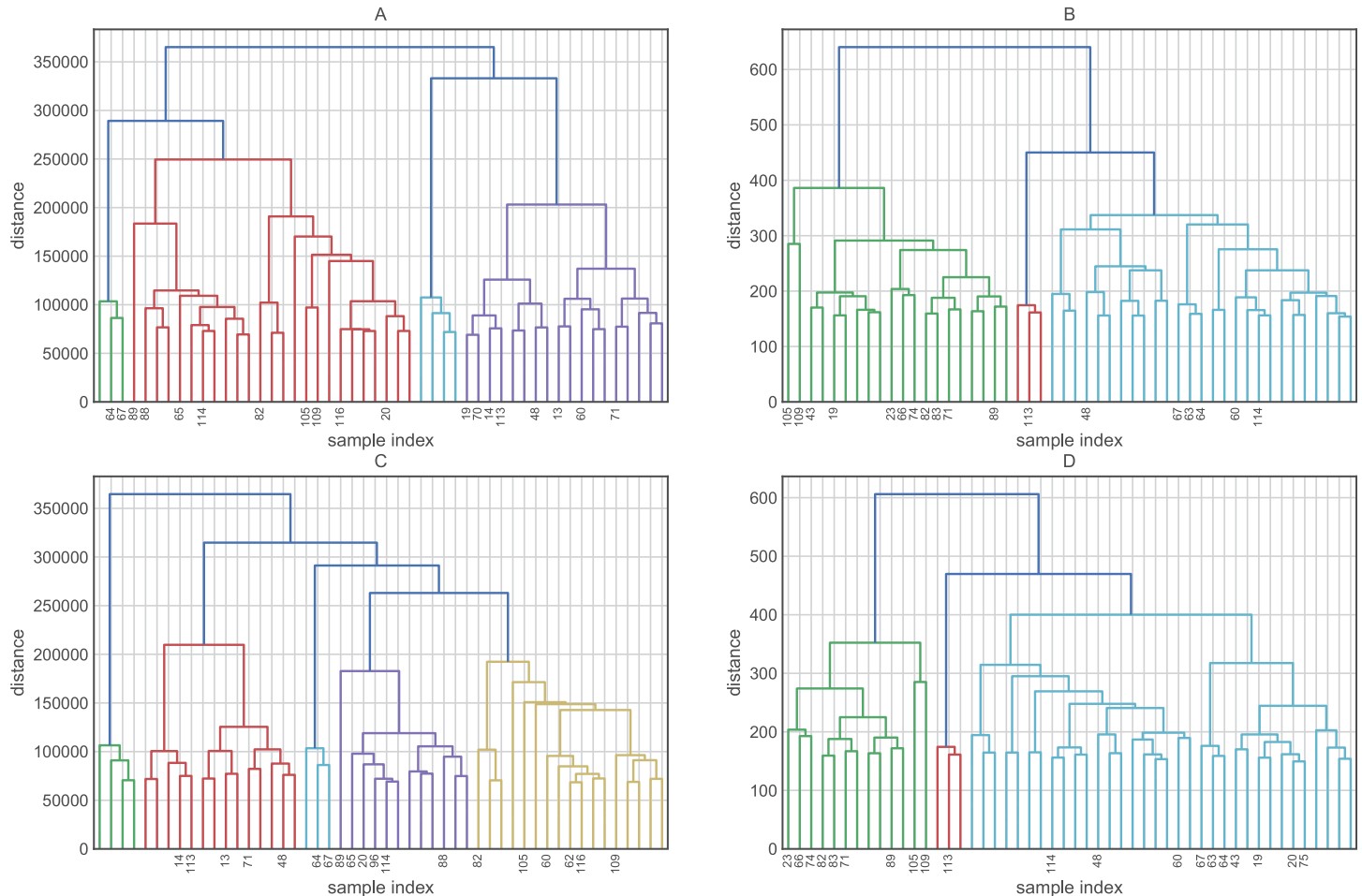

**Figure 2 Hierarchical maps using Ward as the clustering method and (A) raw data (B) scaled data, (C) data reduced by PCA and (D) data scaled and reduced by PCA.** Due to the large number of characteristics of the dataset, it is recommended that you transform the dataset to use only the most relevant and informative variables, which is called the preprocessing step.

concluded that Ward's method produced the most balanced clusters (Fig. 3). Then, using only Ward's method, we performed additional analyses using different datasets, including raw data, scaled data, data transformed by PCA, and data scaled and transformed by PCA. Finally, we created a dendrogram and a heatmap to find whether data can be clustered into groups without any given class with the best results. Figure 4 shows four well-separated groups, but the heatmap demonstrated other well-conserved groups, which may indicate that the four main clusters could be divided into subgroups.

Ward's method created four groups, while the other methods clustered the individuals into fewer groups and, in most cases, these groups are largely unbalanced. On the other hand, the raw data and data transformed by PCA performed better in the hierarchical clustering analysis. Employing these datasets, we were able to obtain four and five clusters, respectively. Finally, the heatmaps plotted in Fig. 4 showed one group greatly distant from the others (green in Fig. 4A and light blue in Fig. 4B). On the other hand, the other

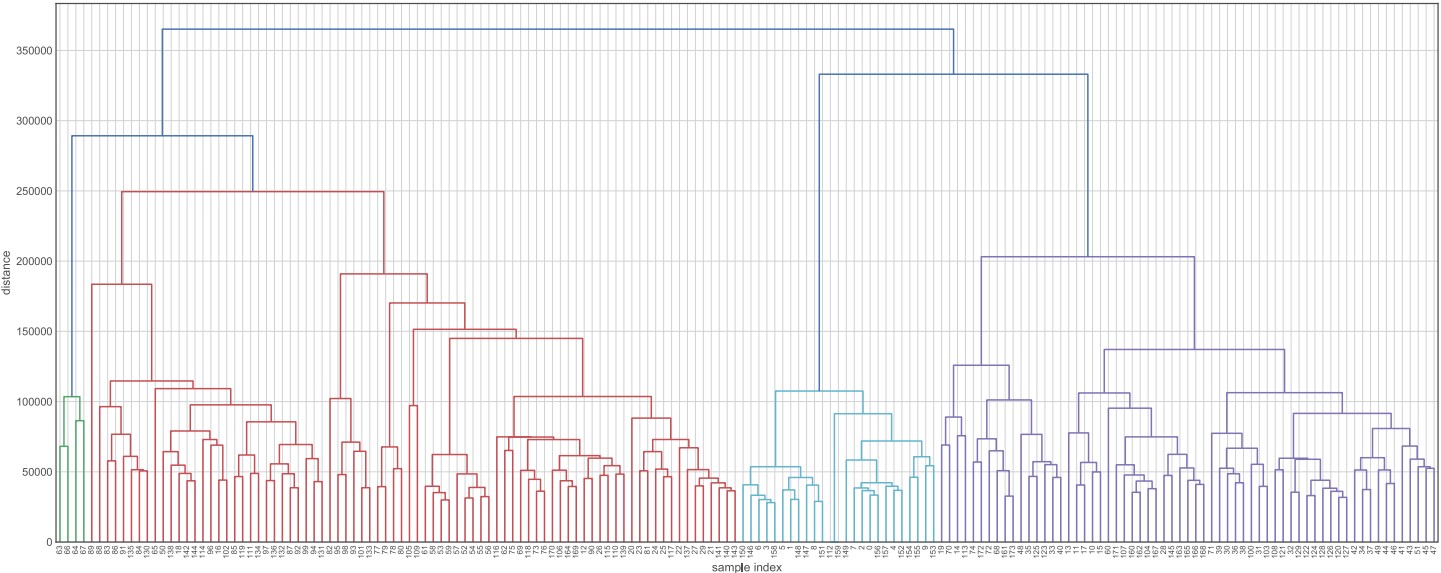

**Figure 3** **Hierarchical maps using Ward's method as the criterion for choosing the pair of clusters to merge at each step.** This hierarchical map was generated by data without transformation and deleting their labels. Clustering approaches demonstrate whether the data contain relevant patterns for grouping.

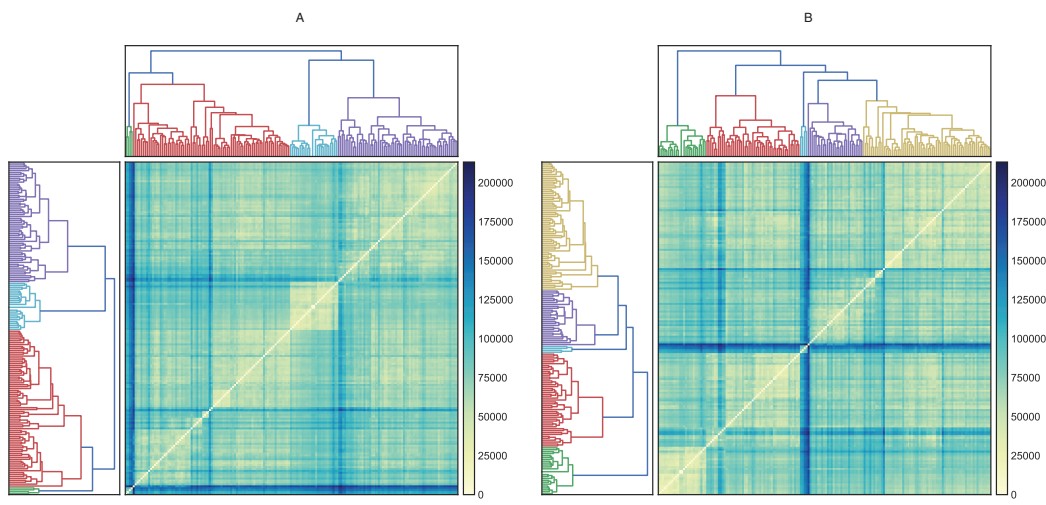

**Figure 4** **Hierarchical and heatmap analysis utilizing (A) raw data and (B) data processed by PCA.** These heatmaps show how similar (near zero) or different (about 200,000) the individuals in the clusters are. A cluster is interesting when its members are very similar and are very different from individuals in other groups.

clusters showed low intra-cluster distances, which is an ideal feature in classification problems (clear blue in Fig. 4A and green in Fig. 4B).

Based on a priori knowledge that the number of cancer types is eleven (11), we were interested in determining how the hierarchical clustering algorithm created the cluster assignments. Therefore, we applied the best parameters found previously (clustering

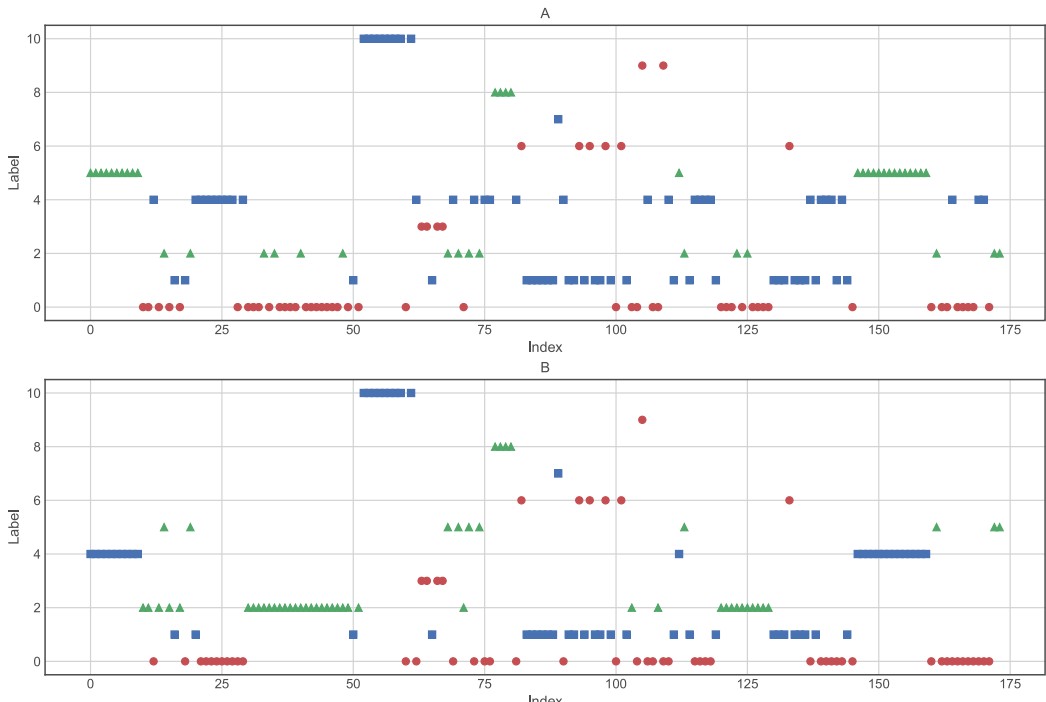

**Figure 5 Clusters composition using (A) raw data and (B) data processed by PCA.** Clustering was performed using Ward as the distance algorithm. Label correspond to the cluster number predicted by the algorithm and may not correspond to labels of Table 1.

**Table 4 Cluster composition and original number of individuals from each class of cancer.**

| Class | Original number | Clustering using raw data | Clustering using data processed by PCA |
|---|---|---|---|
| 0 | 27 | 47 | 47 |
| 1 | 8 | 29 | 28 |
| 2 | 26 | 16 | 39 |
| 3 | 23 | 4 | 4 |
| 4 | 12 | 31 | 25 |
| 5 | 11 | 25 | 10 |
| 6 | 7 | 6 | 6 |
| 7 | 26 | 1 | 1 |
| 8 | 6 | 4 | 4 |
| 9 | 14 | 2 | 1 |
| 10 | 14 | 9 | 9 |

method: ward, and input: raw data and data reduced by PCA). The results shown in Fig. 5 and Tables 4 and 5 demonstrate that, although the hierarchical clustering algorithm displays good performance, it does not group the data into the correct number of groups.

Another unsupervised learning assessment involved the implementation of the *K*-means algorithm. We used all datasets and changed the number of clusters iteratively

**Table 5 Metrics obtained by *K*-means for each cancer type.**

| Class | Precision | Recall | F1-Score |
| --- | --- | --- | --- |
| 0 | 0.74 | 0.68 | 0.71 |
| 1 | 0 | 0 | 0 |
| 2 | 0.45 | 0.9 | 0.6 |
| 3 | 0.68 | 1 | 0.81 |
| 4 | 0 | 0 | 0 |
| 5 | 0.91 | 1 | 0.95 |
| 6 | 1 | 0.4 | 0.57 |
| 7 | 1 | 0.95 | 0.98 |
| 8 | 0 | 0 | 0 |
| 9 | 1 | 0.11 | 0.2 |
| 10 | 0.53 | 0.89 | 0.67 |

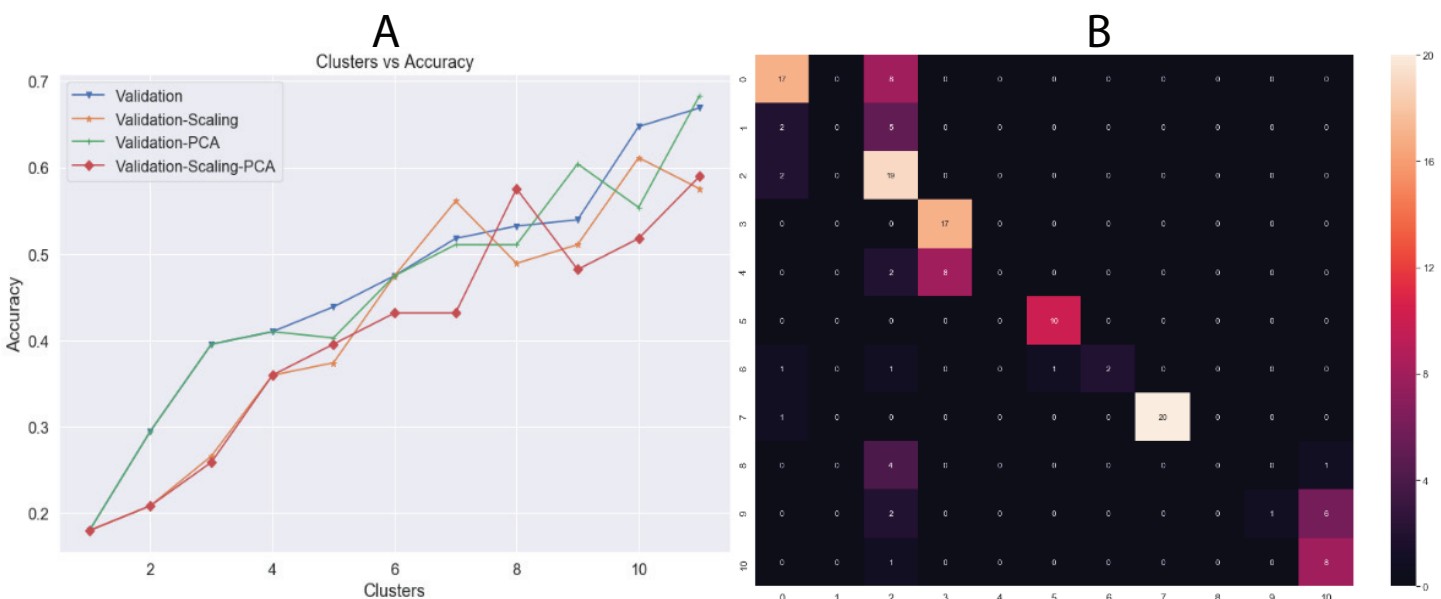

**Figure 6 (A) Behavior of Accuracy in terms of number of clusters and (B) confusion matrix with best results (clusters = 11) using *K*-means algorithm.** Results showed in (A) are the accuracy using validation dataset which correspond to 20% of whole data.

from one to eleven, increasing by one cluster at a time. Then, we calculated the accuracy in each iteration and a confusion matrix was plotted with the best results (Fig. 6). Additionally, we calculated other metrics, such as precision, recall, and f1-score for each class. Overall, the best results were obtained by *K*-means using 11 clusters with input data processed by PCA, achieving an accuracy of 68.34% (validation set, using the hold-out splitting method). Also, classes 6, 7 and 9 showed precisions of 100% and class 5 of 91% (Table 5).

## Algorithm tuning

The algorithms were tuned by setting several parameters between a given value range (Table 2) to find the best behavior using all datasets. Through this, we aimed to calculate the best hyperparameters for each algorithm and determine which dataset could be the most appropriate. The results of the highest validation accuracies are shown in Table 6. To evaluate overfitting or underfitting, we plotted the accuracy values of the training and validation processes on all datasets described above (Fig. 7). RF and DT were not plotted since more than one hyperparameter were tuned. The best results were obtained using LG and raw data. We also calculated a confusion matrix for these results, finding very good classification rates (Fig. 8).

## Cross-validation

KNN, SVC, LG, MLP, K-MEANS, LDA, NB, RF and DT were trained and validated with the same fraction of data and each experiment was repeated 10 times to obtain the standard deviations using sklearn's cross-validation function with $k = 10$ (*Komer, Bergstra & Eliasmith, 2014*). We used the entire dataset (174) for this procedure. The accuracy and standard deviation results are shown in Table 7.

## Deep neural networks

The grid-search method showed the hyperparameter values that provided the best accuracy in FNN and CNN architectures (Table 8). Figures 9 and 10 show the training results of both architectures, demonstrating how the loss function decreases when most epochs are used until a specific number of epochs is reached (80 for FNN and 8 for CNN). Similarly, the accuracy increases in both the training and validation data until reaching the same number of epochs mentioned for the loss function. After this number of epochs, no significant changes were observed for the loss and accuracy values. Using these parameters and cross-validation with $k = 10$, FNN and CNN achieved accuracies of 91.43% and 94.43%, respectively.

## Significance tests

We performed a test of significant differences, with a 95% confidence level, between the two best-performing ML algorithms (LG and CNN). Accordingly, we found no significant differences between the accuracies of these two algorithms ($p$-value = 0.447).

## DISCUSSION

In this work, we show the application of unsupervised and supervised learning approaches of ML and DL for the classification of 11 cancer types based on a microarray dataset. We observed that the best average results using the training and validation data are obtained using the raw dataset and the LR algorithm, yielding an accuracy value of 100% (validation set, using the hold-out splitting method). One could assume there is overfitting since the confusion matrix showed an extremely good behavior; however, the comparison of the training and validation accuracies between parameters using the entire

**Table 6 Tuning hyperparameters of best results of algorithms tested.**

| Algorithm | Conditions on the dataset | Tuning parameters | % Accuracy |
|---|---|---|---|
| Results on validation data (the best result) | | | |
| K-Nearest Neighbors | Any | Neighbors=1 | 88.57 |
| | Scaling | Neighbors: 1 | 71.43 |
| | PCA | Neighbors: 1 | 82.86 |
| | Scaling + PCA | Neighbors: 4 | 48.57 |
| Support Vector Classifier | Any | C=10 | 8.57 |
| | Scaling | C=70 | 94.29 |
| | PCA | C=10 | 8.57 |
| | Scaling + PCA | C=40 | 91.43 |
| Logistic regression | Any | C=0,1 | 100.00 |
| | Scaling | C=0,1 | 97.14 |
| | PCA | C=0,1 | 94.29 |
| | Scaling + PCA | C=0,1 | 94.29 |
| Linear discriminant analysis | Any | Default | 91.43 |
| | Scaling | Default | 91.43 |
| | PCA | Default | 97.14 |
| | Scaling + PCA | Default | 82.86 |
| Gaussian NB | Any | Default | 85.71 |
| | Scaling | Default | 85.71 |
| | PCA | Default | 80.00 |
| | Scaling + PCA | Default | 71.43 |
| Random forest | Any | n_estimators=81, max_depth=91, min_samples_split=10, max_features=50 | 97.14 |
| | Scaling | n_estimators=91, max_depth=81, min_samples_split=10, max_features=60 | 97.14 |
| | PCA | n_estimators=91, max_depth=21, min_samples_split=10, max_features=30 | 94.28 |
| | Scaling + PCA | n_estimators=61, max_depth=11, min_samples_split=10, max_features=20 | 85.71 |
| Decision tree | Any | max_depth=71, min_samples_split=10, max_features=40 | 68.57 |
| | Scaling | max_depth=51, min_samples_split=10, max_features=60 | 68.57 |
| | PCA | max_depth=81, min_samples_split=10, max_features=30 | 82.85 |
| | Scaling + PCA | max_depth=51, min_samples_split=20, max_features=60 | 74.28 |
| Multi-layer perceptron | Any | Neurons=800 | 85.71 |
| | Scaling | Neurons=50 | 91.43 |
| | PCA | Neurons=300 | 97.14 |
| | Scaling + PCA | Neurons=50 | 91.43 |
| K-means | Any | Clusters=16 | 76.97 |
| | Scaling | Clusters=14 | 68.34 |
| | PCA | Clusters=16 | 73.38 |
| | Scaling + PCA | Clusters=11 | 58.99 |

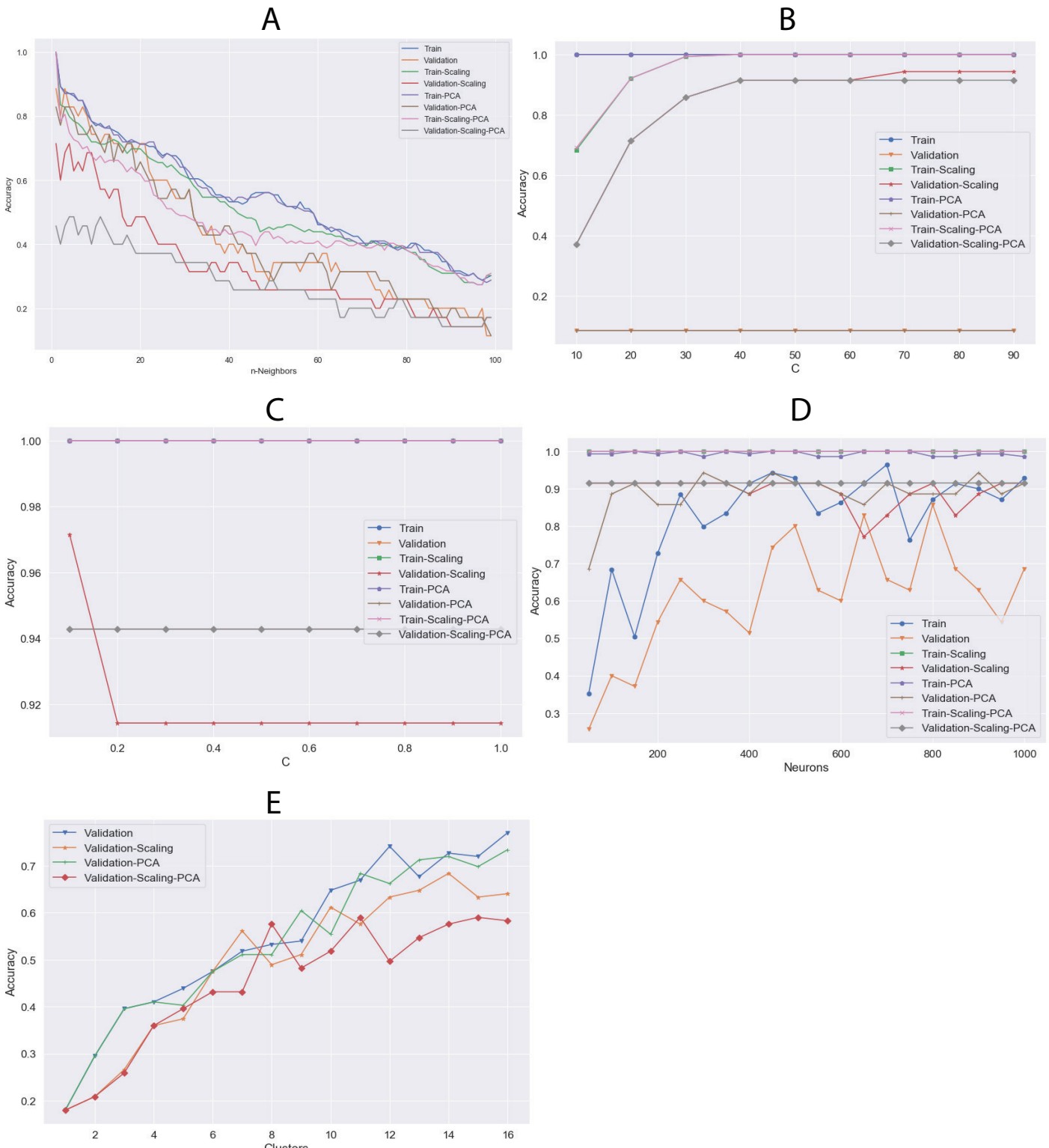

**Figure 7 Comparison of training and validation accuracy between parameters using all dataset and (A) KNN, (B) SVC, (C) LG, (D) MLP and (E) K-means.** The algorithm do not present in this figure; it appears in Table 6 as default in the "Tuning Parameters" column.

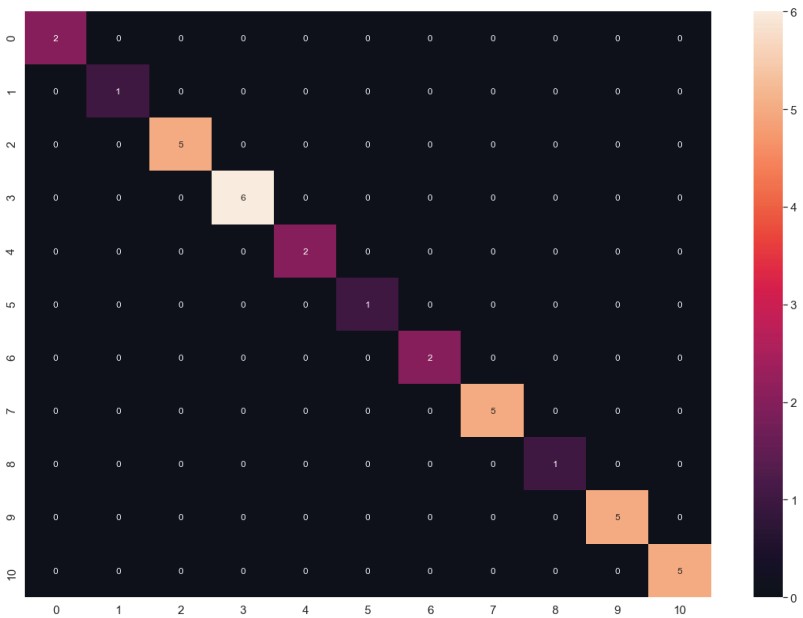

**Figure 8 Confusion matrix of LG algorithm results.**

**Table 7 Cross validation of KNN, SVC, LG, MLP, K-Means, LDA, NB, RF and DT before and after of tuning process.**

| Algorithm | Before tuning | | After tuning | | Significance difference |
|---|---|---|---|---|---|
| | % accuracy | Standard deviation | % accuracy | Standard deviation | |
| Results of cross validation (10 splits) | | | | | |
| KNN | 78.3 | 12.71 | 82.03 | 10.19 | NO |
| SVC | 10.82 | 6.65 | 81.98 | 13.7 | YES |
| Logistic regression | 90.6 | 7.93 | 90.6 | 5.94 | NO |
| Multi-layer perceptron | 79.89 | 20.62 | 83.40 | 13.64 | NO |
| *K*-means | 10.16 | 9.36 | 68.34 | 9.26 | YES |
| Linear discriminant analysis | 83.4 | 11.62 | N/A | N/A | N/A |
| Gaussian NB | 84.12 | 12.78 | N/A | N/A | N/A |
| Random forest | 66.75 | 13.79 | 72.69 | 15.85 | NO |
| Decision tree | 69.78 | 14.9 | 66.04 | 15.45 | NO |

dataset may indicate perfect accuracy in both training and validation datasets. Additional tests with independent data should be done to discard potential overfitting.

On the other hand, MLP and LDA showed a high accuracy value of 97.14% in the validation dataset. This improvement in accuracy was obtained by optimizing several parameters (number of neurons in MLP) and preprocessing the dataset with PCA.

After tuning four parameters, RF obtained high results, with a maximum accuracy of 85.71%. In contrast, DT obtained 51.14% accuracy, demonstrating that DT does not work properly for the datasets used in this study, despite tuning several parameters (in our case, three).

**Table 8 Best value of hyperparameters tuned in deep neural networks.**

| Parameter | Best value | |
|---|---|---|
| | FNN | CNN |
| Batch size | 20 | 10 |
| Epochs | 100 | 10 |
| Training optimization algorithm | Adagrad | SGD |
| Learn rate | 0.2 | 0.1 |
| Momentum | 0 | 0 |
| Network weight initialization | Normal | Glorot_normal |
| Neuron activation function | Softsign | Linear |
| Weight constraint | 3 | 1 |
| Dropout regularization | 0 | 0.4 |

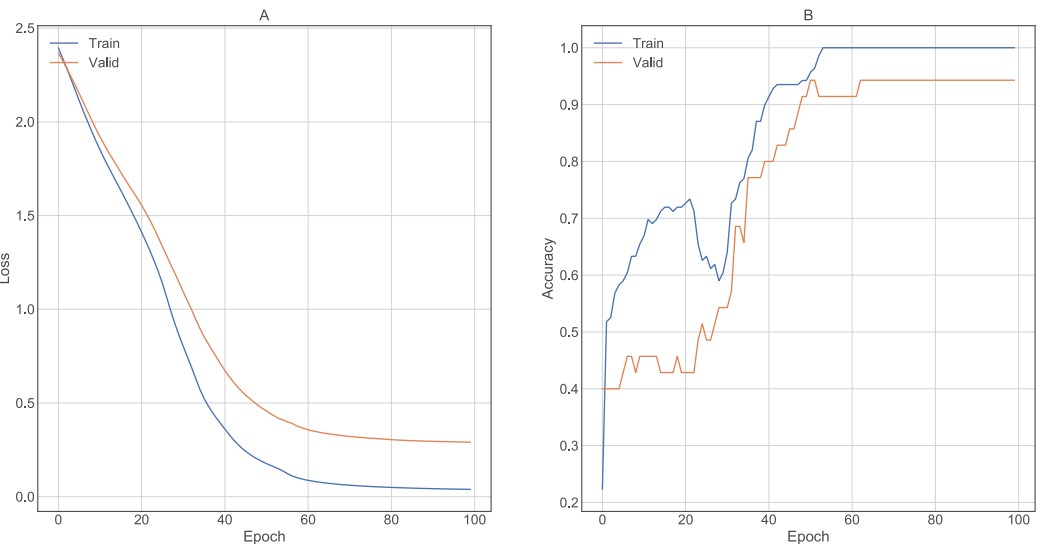

**Figure 9 Results obtained by FNN architecture in training using 100 epochs.** (A) Lost value and (B) Accuracy. Lost function and accuracy is plotted on both training and validation datasets in order to observe behavior. When both datasets show very distant results, the architecture may be overfitting.

Our findings demonstrate that the various algorithms work better by preprocessing the datasets differently. Our results show that MLP, DT, and LDA improved in performance if PCA was applied in advance. However, LG, KNN, NB, RF, and *K*-means worked better using no preprocessing. Only SVC improved when using scaling and, interestingly, none of the other algorithms showed better results using scaling and PCA on the datasets.

Parameter tuning can improve the accuracy of the algorithm used (Table 7). For instance, SVC obtained a low accuracy of 10.82% before preprocessing but increased to 81.98% after tuning. Although most of the algorithms improved their accuracies after the tuning process, only two of them (SVC and *K*-means) showed significant changes. We conclude that LG is the best ML algorithm for the test dataset in this study, providing

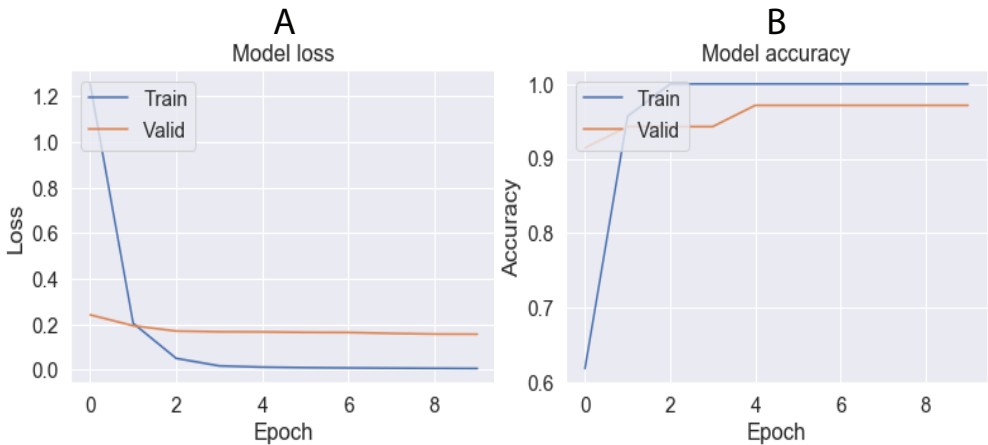

**Figure 10 Results obtained by CNN architecture in training using 10 epochs.** (A) Lost value and (B) Accuracy. Lost function and accuracy is plotted on both training and validation datasets in order to observe behavior. When both datasets show very distant results, the architecture may be overfitting.

an accuracy of 90.6% with a standard variation of 5.94 from the cross-validation analysis based on ten times. Nevertheless, we recommended using it with moderation. On the other hand, for DL architectures, CNN obtained the best accuracy with 94.43% (Fig. 10). The grid search technique enabled parameter tuning and improved the results, allowing us to propose new DNN architectures (i.e., the architectures showed in Fig. 1). Finally, we found no significant difference between the accuracies obtained by LG and CNN.

## CONCLUSIONS

Cancer is predicted to become the most deadly disease for humans in the future (*Dagenais et al., 2019*); therefore, early diagnosis, identification, and treatment are needed to control the disease. ML and DL techniques are promising tools for the classification of cancer types using complex datasets, such as microarrays. In this study, we obtained predictions with as high as 93.52% and 94.46% accuracies, which will allow patients with these types of pathologies to receive an early and precise detection of their disease, and will also contribute to the discovery of new selective drugs for the treatment of these types of tumors.

### Funding

Simon Orozco-Arias is supported by a Ph.D. grant from Ministerio de Ciencia, Tecnología e Innovación de Colombia (Minciencias), Convocatoria 785/2017 and Universidad Autónoma de Manizales, Manizales, Colombia supported and covered the publication fees under the project 589-089. The funders had no role in study design, data collection and analysis, decision to publish, or preparation of the manuscript.

## Grant Disclosures
The following grant information was disclosed by the authors:
Ministerio de Ciencia, Tecnología e Innovación de Colombia (Minciencias), Convocatoria: 785/2017.
Universidad Autónoma de Manizales, Manizales, Colombia: 589-089.

## Competing Interests
The authors declare that they have no competing interests.

## Author Contributions
- Reinel Tabares-Soto conceived and designed the experiments, performed the experiments, performed the computation work, prepared figures and/or tables, authored or reviewed drafts of the paper, and approved the final draft.
- Simon Orozco-Arias conceived and designed the experiments, performed the experiments, performed the computation work, prepared figures and/or tables, authored or reviewed drafts of the paper, and approved the final draft.
- Victor Romero-Cano conceived and designed the experiments, analyzed the data, prepared figures and/or tables, authored or reviewed drafts of the paper, and approved the final draft.
- Vanesa Segovia Bucheli analyzed the data, authored or reviewed drafts of the paper, and approved the final draft.
- José Luis Rodríguez-Sotelo analyzed the data, authored or reviewed drafts of the paper, and approved the final draft.
- Cristian Felipe Jiménez-Varón analyzed the data, performed the computation work, prepared figures and/or tables, authored or reviewed drafts of the paper, and approved the final draft.

## Data Availability
Data is available at GitHub: https://github.com/simonorozcoarias/ML_DL_microArrays.

## Supplemental Information
Supplemental information for this article can be found online at http://dx.doi.org/10.7717/peerj-cs.270#supplemental-information.

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
