# Peer review of "A comparative study of machine learning and deep learning algorithms to classify cancer types based on microarray gene expression data"

_PeerJ Computer Science, doi:10.7717/peerj-cs.270_

## Round 0.1 · original submission · Major Revisions

The authors should clearly explain the novelty of their work, if any. This includes the motivation for the experiments and a better discussion of the obtained results. The authors should also provide a working link to the dataset.

Reviewer 1 ·

Basic reporting

The authors compare many different clustering and classifier algorithms for identifying 11 types of cancer on a microarray dataset. Very few references are given to the previous applications of machine learning to microarray data, which is a vast literature. The exact contribution of the study, compared to other works, is also unclear. Many authors have compared different machine learning techniques on microarray data. Deep learning has also already been used. What exactly is new in this study? The main motivation provided in the abstract is that there have been "advances in both hardware and software technologies", which is not enough to motivate the analysis. I have also some concerns regarding how the experiments were carried out (see below). Finally, the language needs some improvement. Due to the aforementioned reasons, I do not recommend publication in PeerJ Computer Science unless there are some major improvements to the manuscript.

- Most of the introduction is spent talking about Deep Learning, supervised vs unsupervised learning and machine learning in general. Prior work on machine learning methods are note discussed, only briefly mentioned. The authors should include a literature review section describing previous works in order to provide context for their study.

- Almost no description is given about the dataset. What does the data represent (besides gene expression)? What is each of the 174 samples? What is the number of objects in each class (this is only provided in table 4 in the results)? Is this data unique in some aspect? Besides, the provided link (http://www.gems-system.org) is not working.

- The authors should provide a brief explanation about each algorithm. Some algorithms (and parameters) are more suited for unbalanced classes, for high dimensional data, etc. The authors need to provide some prior information on what is expected from each algorithm for this particular data.

Experimental design

- The authors mention that they used a training, validation and test set. It is not clear if the test set was used only after tuning all parameters of the methods on the training and validation sets. Actually, looking at the source code it is not clear that a test set was used at all. If the parameters were tuned and the accuracies were quantified using the same data, even with cross-validation, there will be overfitting. The test set should be set aside and not be used in any of the training/parameter tuning steps. They are used for testing the final performance of the methods only after tuning all parameters in the training/validations sets. Alternatively, a nested cross-validation scheme might be used (one for measuring the accuracy during training and another for the final accuracy after parameter tuning).

- The author mention that "the Ward method is the one with the best performance". This is based on which criteria? The Ward method can find "well-separated" clusters even in normally distributed data. Just because the method managed to find clusters it does not mean that they are the correct clusters, or that the data has any cluster at all.

- Why are CNNs being used? Do the order of the rows and columns of the gene expression table have a special meaning? In my experience with this type of data, usually do not have any particular meaning, that is, either the columns or rows can be shuffled without loos of information. If there is no spatiality in the data, it is difficult to see why CNNs are useful here.

Validity of the findings

- See above for the validity of the findings, which depends on the experimental design.

Additional comments

- The authors wrote

"of each algorithm to find the “real” accuracy. Additionally, "

Please define what is the meaning of "real".

- Figure 2 is mentioned before Figure 1

- The DL architecture used is only presented on the last paragraph of the results section. It should be placed before the results.

- Many English mistakes in the text. Some examples:

- since allows an improved

- and guaranteeing the effectiveness and less side effects

- ML and DL techniques are entitled to learn characteristics of

- using K-means algorithm varying amounts of cluster between one and eleven using all datasets

- a sequential process composed by choosing values of parameters which achieved best accuracy and then used it to found others

- except RF and DT due to more than one hyperparameters were tuned.

Reviewer 2 ·

Basic reporting

The article presents some problems with the English language. There are wrong or incoherent expressions in many parts of the article. These wrong expressions need to be edited, and they can be found as follows:
- Abstract: lines 22, 24, and 33
- Introduction: lines 40, (42,43,44), 65, 72, and (74-75)
- Preparing the data: lines (118,119)
- Unsupervised learning experiments: lines 123, 132, and 134
- Results: 172, 197, (213,214), (218,219)
- Conclusions: 288
- Figures 2 and 6: caption

It would be interesting to explain a little better the motivations of the authors for the task of cancer types classification; in my opinion, the explanation was a bit simple. Also, you should mention the importance of using machine learning and deep learning methods.

You described a few related works about the task you are researching. You should mention more works that developed the task of cancer types classification for both machine learning and deep learning methods. Also, you should mention more works that have used the "11_tumor" database.

Experimental design

You should explain with more detail the database you are using (11_tumors database), maybe with some examples of how a gene is represented in such a database.

In the unsupervised learning experiments section, at line 131, you should explain a little each of these metrics, but the most relevant ones. It would be important to include some references.

In tuning the algorithms section, at lines 155 to 158, you should improve the writing and explanation of the steps for finding the best value of parameters

Validity of the findings

no comment

Additional comments

The article is very interesting and the authors developed many experiments and results for the task of cancer types classification. It was relevant that you addressed several machine learning approaches like supervised and unsupervised classification, and also deep learning methodologies. However, there are some mistakes that need to be taken into account and corrected so that the article can be accepted. These issues are detailed in the Basic Reporting and Experimental design sections.

·

Basic reporting

The authors present a comparative study of several supervised, unsupervised and deep learning algorithms to understand which the best algorithms are to deal with microarray 11 tumor database.

The article is well written but the introduction needs more details. The related works that were described in the introduction were superficially approached and many of them had no direct relation to the manuscript research. I suggest creating a new section of related work.

Experimental design

There are related works such as Machine Learning in DNA Microarray
Analysis for Cancer Classification who makes an analysis similar to the analysis presented in the manuscript. What is the contribution of this manuscript to others?

No statistical analysis was mentioned in the manuscript.

Why was chosen PCA instead of applying feature selection techniques? What were the motivations that led to this decision?

What was the reason for choosing the algorithms that were used in the comparative analysis? The text did not mention the reason.

Validity of the findings

The link provided to access the database does not contain the database. The authors should verify that the database is publicly available.

The text describing the result (page 11) is not well detailed. For example, indicate why some techniques obtained a high accuracy when applying PCA and scaling and others not.

---

## Round 0.2 · Major Revisions

Some issues have been addressed, however, there are some major issues that still needs your attention. Particularly, the authors should address (or respond accordingly) the overfitting issue. A significance test is also expected.

Reviewer 1 ·

Basic reporting

The authors answered all of my queries. Therefore, I consider the manuscript suitable for publication in PeerJ Computer Science.

Nevertheless, there are many grammar mistakes remaining in the text.

Experimental design

No comment

Validity of the findings

It is still unclear if the CNN is overfitting to the data, since the hyperparameters were tuned on the same set that was used for accuracy calculation. Still, the results show that the model have enough capacity to reach high accuracies.

Additional comments

No comment

·

Basic reporting

In the earlier version of the manuscript, the main contribution of the work was missing. The lack of an adequate bibliographic review describing the related works to contextualize the problem and the poorly detailed way in which the study was approached decreased the relevance of the work, no showing any kind of contribution. In this new version of the manuscript, it is possible to see the contribution of the work indicated in paragraph 3 of page 6 and the last paragraph on page 7. Although the contribution of the work is now visible, the authors have very difficult detailing their explanation. These issues are detailed in the Experimental design sections.

Experimental design

Next, I will explain some issues related to the text: For example, in paragraph 3 of page 6, it is mentioned by the authors that several other works applied machine learning techniques to the 11-tumor dataset, but they only used some machine learning methods, one preprocessing strategy and one learning technique (supervised or unsupervised). The information is very vague. Should be mentioned these methods. What preprocessing technique was applied? What unsupervised learning algorithms the authors use? What did these authors conclude? Is there any relationship between the results obtained from this manuscript and the other works, or not? All of this information is relevant because it will validate the study done in this work.

Validity of the findings

When a comparative analysis is done, it is expected to see the statistical significance of comparing the algorithms used. For example, if one algorithm had an accuracy rate of 90% and another 92%, we could say that this difference was statistically significant or not? I was hoping to find this statistical analysis in the text.

Additional comments

1. As there were several changes in the text, the abstract must be updated to reflect the clarifications made by the authors on the contribution and the results.
2. About my question of why a feature selection method had not been used I understand that feature selection was not considered by project scope. Only I want to clarify to the authors that PCA is not a feature selection, is a dimensionality reduction algorithm that performs a linear transformation of the data into a lower dimension. On the other hand feature selection select and exclude features keeping the features intact with any kind of transformation.

---

## Round 0.3 · accepted · Accept

All issues have been addressed and therefore your manuscript can be considered acceptable for publication.

Reviewer 2 ·

Basic reporting

no comment

Experimental design

no comment

Validity of the findings

no comment

Additional comments

All previous suggestions were covered, therefore I consider the manuscript suitable for publication in PeerJ Computer Science.

·

Basic reporting

No comment

Experimental design

No comment

Validity of the findings

No comment

Additional comments

No comment